# The Relationship between Modern Fad Diets and Kidney Stone Disease: A Systematic Review of Literature

**DOI:** 10.3390/nu13124270

**Published:** 2021-11-26

**Authors:** Yazeed Barghouthy, Mariela Corrales, Bhaskar Somani

**Affiliations:** 1GRC n°20, Groupe de Recherche Clinique sur la Lithiase Urinaire, Hôpital Tenon, Sorbonne Université, F-75020 Paris, France; yazeedmail@gmail.com (Y.B.); mariela_corrales_a@hotmail.com (M.C.); 2Department of Urology, University Hospital Southampton NHS Trust Southampton, Southampton SO16 6YD, UK

**Keywords:** diet, fad diets, low carbohydrate, high protein, low fat, Atkins, zone, ketogenic, Dukan, Mediterranean, vegetarian, vegan, paleo, nephrolithiasis, urinary stones, kidney stones, urolithiasis, risk, kidney calculi

## Abstract

Objectives: Kidney stone disease (KSD) has a strong association with diet metabolic syndrome. This review aims at exploring the lithogenic risk posed by the current most popular diets. Our approach was to search for the effect of each diet type on the major urinary risk factors, to try to draw conclusions regarding the association of a specific diet type and KSD. Methods: This systematic review searched for the available literature exploring the association between the existing popular fad diets and KSD. Articles in English, French and Spanish were included, without restriction of the search period with the final search done in August 2021. Results: Total number of studies and studies for each diet type was as follows: 22 articles for the low carbohydrate diet, 20 articles for high protein diets, 26 articles for vegetarian and vegan diets. There exists a substantial variability in different low carbohydrate and high protein diets, and considerable overlap between modern popular fad diets. High carbohydrate intake might increase urine uric acid, calcium and oxalate levels. High protein diets increase urine calcium and uric acid and lower urine pH and citrate. Consumption of fruits and vegetables increases the urinary volume and urinary citrate. In vegan diets, sufficient daily calcium intake is important to avoid possible secondary hyperoxaluria. Conclusions: Few studies evaluated the direct relationship between modern fad diets and KSD. In general, the reduction of carbohydrate in the diet, and counterbalancing protein rich diets with sufficient intake of fruits and vegetables, seem to play a protective role against KSD formation. Maintaining sufficient calcium intake in vegan and vegetarian diets is important. Additional research is needed to directly evaluate the link between KSD and each diet type.

## 1. Introduction

Kidney stone disease (KSD), with an estimated prevalence of up to 10–14% in industrialized nations [1,2], is closely related to obesity and other components of the Metabolic Syndrome (Mets) [3]. The increase in prevalence of this ensemble of diseases in recent decades is attributed mainly to the changes in lifestyle habits and dietary intake, notably the significant increase in carbohydrate and protein intake [4,5,6,7,8].

Kidney stones disease is a multifactorial process, influenced by dietary, urinary, and genetic factors. The early steps involve mineral precipitation secondary to reduced fluid intake and over-saturation of their urinary levels, followed by crystal nucleation, aggregation, and growth to particles measuring a few hundred microns. In turn, these particles might agglomerate into small stones that continue to grow in the renal collecting system, in different rates, according to the underlying pathology. While all stones share the common pathologic pathway of a disequilibrium between promoting and inhibitory factors for stone formation, each family of stones has a more characteristic set of risk factors. Calcium stones, whether associated to oxalate or phosphate, represent the majority of stones. They are influenced by excessive urine calcium and oxalate levels, due to high dietary intake or intestinal malabsorption syndromes, or a rise of urine calcium due to parathyroid disease. Uric acid stones on the other hand, which constitute almost 10–15% of stones, are closely linked to high dietary protein and uric acid intake, and to metabolic acidosis in addition to insulin-dependence.

Our diet is composed mainly of three major nutrients—carbohydrates (4 kcal/gm), protein (4 kcal/gm), and fat (9 kcal/gm), with the recommended composition range being 45–55%, 10–35%, and 20–30% respectively, accounting for around 35 Kcal/Kg per day [9]. While multiple definitions exist for the term “Fad diets”, the common ground for all these definitions is that fad diets are popular “weight loss” regimens, promoted throughout popular media outlets and sometimes endorsed by celebrities. They usually focus on manipulation and elimination of certain nutritional components rather than larger life-style and health modification based on scientific research. Furthermore, fad diets in many cases are economic tools used to promote sales of books or products such as “weight-loss” programs. Three main categories of diets are available, those based on the manipulation of macronutrient content such as low carbohydrate diets or protein rich diets; those based on the restriction of specific foods and/or food groups such as the vegan or vegetarian diets; and, finally, diets based on intermittent fasting such as the “moon diet”. In the last twenty years, a number of fad diets have experienced growing popularity, sometimes under the influence of social media and popular trends. The common goal of these diet strategies, in addition to losing surplus weight, is to reduce the risks for cardiac disease and diabetes mellitus [10]. While some studies showed an advantage for weight loss in the short term for mainly for protein rich low carb diets, the results for the best strategy for weight loss in the long term are a matter of debate. Moreover, fad diets demonstrate the social pressure for achieving an idealized body image through control of nutrition [11].

The co-existence of KSD and obesity, and the prescription or self-adherence of certain diets to combat obesity, leads us to inquire about what effect these different diets would have on urinary risk factors for stone formation. However, the real impact of these fad diets, on the risk of kidney stones’ formation, has not been thoroughly evaluated. Furthermore, despite not being the focus of this paper, it should be noted that diets involving changes in macronutrient composition or acquiring a state of negative energy balance might cause hormonal and metabolic changes in the body, affecting multiple organ systems. For example, diets refraining from dairy products intake might expose patients to reduced mineral density pathologies, and diets involving intermittent fasting can lead the body to a ketogenic metabolic mode, which are not suitable for all patients [10]. In addition, adhering to a diet without proper information, guidance and support can lead to different psychological perturbations including body-weight image disturbances.

This review aims at exploring the lithogenic risk posed by the current most popular diets. Nevertheless, numerous types of diets have evolved and others have been modifications of already existing ones. In addition, there exists a considerable overlap between different families of diets, as in the case of carbohydrate low and protein rich diets. These factors make it difficult to evaluate the direct specific relationship of each diet with KSD.

In consequence, our approach was to search for the effect of each diet type on the major urinary risk factors, in order to try to draw conclusions regarding the association of a specific diet type and KSD.

## 2. Methods

This systematic review searched for the available literature exploring the association between the existing popular fad diets and KSD. Articles in English, French, and Spanish were included, without restriction of the search period with the final search done in August 2021. Systematic reviews, case reports, editorials letters, comments, and abstracts- were excluded.

### Search Strategy and Study Selection

A systematic review was performed, using the PubMed-MEDLINE, EMBASE and Scopus databases. The Preferred Reporting Items for Systematic Reviews and Meta-Analyses (PRISMA) statement (Figure 1) was followed in this review. The keywords used were the followings: “Diet”, “fad diets”, “Low carbohydrate”, “ketogenic”, “Low Fat”, “Atkins”, “Zone”, “Ketogenic”, “Dukan”, “Mediterranean”, “Vegetarian”, “Vegan”, “Paleo”, “Moon”, and “South beach”. Search terms included a combination of keywords above with each of the following: “Nephrolithiasis”, “Urinary stones”, “Kidney stones”, “Urolithiasis”, “Risk”. Two reviewers (Y.B., M.C.) identified all the studies independently and any discrepancy were resolved with mutual consensus. A narrative synthesis rather than a quantified meta-analysis of data was performed, due to the heterogeneity of outcomes.

## 3. Diet Types and KSD

### 3.1. Low Carbohydrate (LC) Diet

Low carbohydrate (low carb, LC) diets are one of the oldest known methods for weight reduction. Multiple modifications of the LC diet exist, with all entailing a reduction of carbohydrates intake in the diet to less than 45–50% of the total macronutrient intake [6,12].

The common theme to all LC diets is that a lower carbohydrate intake will eventually reduce the levels of circulating insulin levels, switching the body’s metabolism away from the anabolic state, in what constitutes the carbohydrate-insulin model, and eventually inducing a relatively rapid weight loss within the first year [13,14].

To compensate for the reduction of carbohydrate intake in the diet, there is an increase in the percentage of protein or fat intake per day [15,16]. The various versions of LC diets differ in the percentages of carbohydrate intake reduction, and protein or fat increase, and it is from this increase of the other macronutrients that the overlap with other types of diets exists.

The Atkins diet, probably one of the most popular diets in recent decades, is very low in carbohydrates, with an induction phase of 2–3 months with less than 20 g per day of carbohydrate intake and subsequent balancing and maintenance phases with less than 50 g per day of carbohydrate intake. The Zone diet includes a mild reduction in carbohydrate intake to 40% of total macronutrient intake, and a mild increase of protein and fat intake to 30% each of the total nutritional intake per day. The South Beach diet, a more recent approach, is a fiber-rich diet that promotes intake of complex carbs, or so-called good carbs, from various sources such as fruits, vegetables, whole grains, and beans, while avoiding simple or “bad” carbs, represented by sugar and industrial syrups and sweeteners [10,16,17,18]. An extreme modification of a LC diet, the ketogenic or “keto” diet, induces a nutritional ketosis and thus production of ketone bodies from fat tissue in the body, by increasing energy intake from fat, and significant restriction of carbohydrate intake to 20–50 g per day and subsequent glycogen storages’ depletion. This diet is mainly used in the management of refractory cases of epilepsy [19].

### 3.2. LC Diets and Urinary Risk Factors for KSD

#### 3.2.1. Calcium

Previous research, backed by large epidemiological studies, have demonstrated that an oral load of simple carbohydrates, such as glucose or xylitol, can induce hypercalciuria and increase the risk for KSD [20,21,22] (Table 1). It is worth mentioning that the demonstrated hypercalciuria is not attributed to the secretion of insulin in response to sugar intake [23].

This logically leads to the assumption that low carb diets would reduce the risk of hypercalciuria for new stone formation. This positive effect of low carb diets, however, is counterbalanced by the increase in protein intake, to compensate for the necessary caloric requirements. This increase in protein can be a factor for hypercalciuria as will be detailed in the next section dealing with protein rich diets. Nonetheless, even with this effect of hypercalciuria due to the increased protein intake, Friedman et al. demonstrated that no new kidney stone formation was associated with a low carb, protein rich diet [24].

#### 3.2.2. Oxalate

Fructose (largely known as “fruit sugar”) and xylitol (a sugar substitutive and added sweetener) are the two sugars demonstrated to influence oxalate synthesis via the glyoxylate pathway. Previous works, mainly by Nguyen et al. in the 1980s and 1990s, showed that dietary intake of carbs and substituents like xylitol resulted in hyperoxaluria [20,21]; however, a review by Holmes et al. suggested that the consumed amounts that increase oxalate synthesis are not likely to be found in a normal diet [25].

#### 3.2.3. Uric Acid

One of the most important, yet largely overlooked, is the connection between carbohydrate intake in the diet and the rise of uric acid levels with the subsequent risk for uric acid stones. The connection lies with the fact that fructose metabolism significantly promotes ATP degradation to AMP, a precursor of uric acid production, and this can further promote purine synthesis and elevation of uric acid levels [26,27].

Indeed, one of the roots of the obesity epidemic is the introduction in 1967 of the high fructose corn syrup (HFCS), the use of which was popularized by an aggressive marketing campaign, in addition to certain “advantages” in comparison to the traditional cane sugar production, mainly in domains of transportation and manufacture. The use of HFCS in soda beverages was responsible for the even more dramatic increase in its consumption by all segments of the society, adults and minors alike, with a tremendous increase in the daily caloric intake of consumers [28,29].

Large epidemiological studies such as the NHANES III study and the HPFS study both showed that fructose ingestion is associated with increased serum uric acid levels [30,31].

In a study evaluating the effects of fructose ingestion on risk factors for KSD, Johnson et al. evaluated 33 healthy adults after ingestion of fructose daily for two weeks. The results showed an increase in blood uric acid, an increase in urine oxalate, and a decrease in urinary pH and magnesium [32]. The results also showed a trend for a higher level of urine uric acid and a lower level of urinary citrate.

All of the above drives one to the logical assumption that reduction of carbohydrate intake in the diet may avoid increases in urine calcium, oxalate and uric acid levels. However, a uniform conclusion for all low carb diets regarding the risk to develop KSD is not possible. This is mainly due to the variability among different low carb diets under the same umbrella, in a matter of carbohydrate percentage within the diet, in addition to the fact that any benefit of carbs reduction in the diet may be counterbalanced by an increase in protein consumption and the resultant hypercalciuria. The increase in fruits and vegetables’ intake, in order to consume “good” carbs instead of “bad” carbs, probably plays a protective role against urinary stone formation by increasing urinary citrate and dietary fibers. One major issue of concerns, however, is the long-term safety of very low-carb “keto” diets. Sampath et al. suggested that “keto” diets, mainly used in the long term for therapeutic purposes in drug refractory epilepsy in children, can increase the risk of KSD due to hypercalciuria [33]. Maintaining sufficient fluid intake for adequate urine production, and urine alkalinization, in addition to periodic renal ultrasound to screen for stones, are all recommended measures [33,34].

### 3.3. High Protein Diet

Intake of animal protein in the diet has significantly increased in the last several decades with the adoption of a western industrial diet in growing parts of the world. The recommended daily intake of protein in the diet per day is 0.8–1 g protein/kg body weight [35], corresponding to around 15–25% of daily caloric intake. The source of our dietary protein can be animal protein (meat, fish, poultry, eggs) or dairy proteins, in addition to proteins from vegetables. The most popular protein rich diet is the Dukan diet, a high protein, low carb and low fat diet [36]. In addition, many low carb diets include high protein intake, to compensate for the reduced caloric intake from carbohydrates [10]. As will be explored below, previous studies have suggested that a high protein intake increased the risk for urinary stone formation due to its effects on urine calcium, uric acid, oxalate and citrate [22,37,38], while a low protein diet played a protective role [39].

#### 3.3.1. Calcium

A high animal protein diet can significantly increase the acid load on the body [40]. In addition, protein rich diets induce an increase in urine calcium excretion, lower urine pH and lower urine citrate levels [41,42]. This has also raised concerns regarding increased bone reabsorption and possible resultant osteoporosis, even in younger adults. However, later studies suggest that, despite the resultant hypercalciuria, protein rich diets do not seem to influence body calcium balance and do not lead to calcium bone loss [43]. In fact, the hypercalciuria might be the result of increased intestinal absorption of calcium, and some authors suggest that long-term high-protein intake might even increase bone mineral density, through increased intestinal calcium absorption, reduced levels of parathyroid hormone, and higher circulating levels of IGF-1, an anabolic hormone that increases bone mass [44].

#### 3.3.2. Uric Acid

Diets with high intake of purine rich sources (e.g., animal meats, fish, seafood) increase the levels of uric acid levels considerably, thereby elevating the risk for uric acid stone formation. In a small study of 10 male adults, intake of a vegetarian diet rather than a diet containing animal protein, reduced uric acid crystallization by 93% [45]. The source of dietary protein needs to be considered as well, as fish lead to a higher level of uric acid in the urine than animal meat protein, for example [46].

#### 3.3.3. Oxalate

The relationship between protein rich diets and oxalate metabolism is not as extensively evaluated as that for calcium. Previous studies have suggested that, in normal subjects with a high intake of dietary proteins, the catabolism of amino acids did not increase endogenous oxalate synthesis [47]. Another study demonstrated that a protein rich diet increased urinary oxalate excretion in calcium oxalate stone formers but not in controls [48].

#### 3.3.4. Citrate

Breslau et al. demonstrated that citrate excretion in the urine was reduced with increased animal protein intake, probably due to the associated acid load and an increase in renal tubular reabsorption [49].

In conclusion, a high protein intake in the diet increases net acid load, lowers urine pH and citrate excretion, in addition to an increase in uric acid levels. It is also associated with an increase in urine calcium excretion. However, if protein rich diets are associated with sufficient vegetables and fruits intake, the deleterious effect mentioned above can be counterbalanced and the risk for urinary stone formation can be reduced [50]. Moreover, the source of protein in the diet, whether animal/dairy or plant protein, needs to be taken into account. In a recent large study, for example, Shu et al. suggested that plant-based proteins did not increase the risk for stone formation in comparison with animal-based proteins [51].

In addition, another high protein diet worth mentioning, due to its recent popularity, is the Paleolithic or simply “Paleo” diet-. In this diet, aimed to mimic our hunter-gatherer ancestors in the pre-agricultural era, all processed food like cereals and dairy products are excluded, and instead, only unprocessed food like meat, eggs, oils, nuts, and fresh fruits and vegetables are included [52]. The concept behind this diet is that our genome is better adapted to the diet our ancestors adhered to for millions of years of development before the agricultural revolution in the last 10,000 years. This diet might indeed be advantageous in terms of weight loss and control of risk factors for the metabolic syndrome [53]. Regarding urinary stone risk factors, the intake of fresh fruits and vegetables is supposed to produce and net alkaline load to counterbalance the increased protein intake. However, it is the absence of sufficient calcium intake in the diet that is a source of concern [54], which might put the subject at increased risk of calcium oxalate stone formation due to the potential increased hyper-oxaluria [55].

### 3.4. Vegetarian Diets

Even if there are different typologies of vegetarianism [56], all vegetarian diets are oriented towards the intake of legumes, fresh and uncooked fruit, soy products (i.e., tofu), nuts, seeds, and whole grains, while abstaining from eating meat or poultry. However, sometimes their diets may include fish and dairy products [56,57].

The reason why most vegetarians take this path and avoid meat products are numerous—for instance, a deep concern about animal care and welfare, religious and cultural beliefs and personal health [56,58,59]. Many studies have demonstrated its relationship with weight loss [60,61,62,63,64,65] and a decrease in abdominal subcutaneous and visceral fat [62,66]. Moreover, vegetarian diets are of interest in the diabetic population due to the increased insulin sensitivity they offer, more than the one offered by the conventional diabetic diet (isocaloric) [66]; and in the metabolic syndrome population because of the reduction in total cholesterol [66] and low-density lipoprotein (LDL) [62,64,66]. In patients with hypertension, it has been demonstrated that this type of diet improves cardiovascular parameters [67,68] and reverse coronary artery disease [67]. However, it is important to note that, despite having all those beneficial aspects mentioned before, these diets are also related with an increased urinary oxalate excretion, compared to a normal diet [69]. Oxalate-rich diets, such as vegetarian ones, increase the intestinal absorption of oxalate and can cause hyperoxaluria [70].

It is important to note that oxalate absorption is also influenced by other factors, mainly nutritional ones. For instance, both calcium and magnesium can bind to the oxalate to form an insoluble complex in the gut that is eliminated in the feces, limiting its absorption and urinary excretion [71,72]. Another one is the dietary phytic acid, which reduces the intestinal absorption of calcium by binding to it [73]. Thomas et al. observed that the mean intestinal oxalate absorption rate and the mean urinary excretion rate increased only in those who had a low-oxalate vegetarian diet (70 mg oxalate) by 72% and 30%, respectively, compared to those who followed a standard mixed diet (60 mg oxalate) [74]. However, the high-oxalate vegetarian (300 mg oxalate) diet had a similar mean urinary oxalate excretion to the mixed diet [74]. The reason for this can be explained by the role other nutritional components play, creating a balance.

All these factors should be taken into count and controlled by individuals going on a vegetarian path. Especially in calcium oxalate stone patients, this diet can only be recommended if it contains the suggested dietary amounts of calcium and magnesium, given at the time of oxalate ingestion, and if it excludes food components that chelates dietary calcium, as the phytic acid [74]. Adequate calcium intake per day needs to be maintained because a reduction from the standard 1200 mg/day can increase the oxalate dietary absorption by fivefold [72].

Additionally, a balanced vegetarian diet with an adequate fluid intake and a high alkali-load with fruits and vegetables make it possible to have an alkali urinary pH, helping to reduce the risk of uric acid crystallization compared to the regular meat-based diets and therefore decrease the risk of urid acid stones [45].

In summary, a balanced vegetarian diet, which concerns a high intake of fruits and vegetables and low intake in animal proteins, low-fat dairy products, and salt, may help to reduce the risk of stone kidney disease [75,76].

### 3.5. Vegan Diets

In contrast to vegetarian diets, vegan diets exclude animal ingredients in general, animal products (i.e., eggs, dairy, honey), garlic, onion, spring onion, scallions and leeks, and some sugars that used bine char in their manufacturing (i.e., cane sugar) [57]. This type of diet is rich in oxalate, coming from the vegetables and grains, and poor in calcium, due to its exclusion of dairy products [75]. These two features may relate vegan diet with calcium oxalate stone formation.

Furthermore, dairy products have been demonstrated to have lower circulating concentrations of urate after its consumption and also to increase uric acid excretion [77,78]. Therefore, in vegan diets, the lack of dairy products is associated with higher serum uric acid, hyperuricosuria, and uric acid stones [79]. Uric acid levels are higher than the one found in meat eaters, fish eaters, and vegetarians [80].

The real effect that this diet has in urinary stones formation is still under investigation and needs further study. However, because of the potential risks that it poses for increasing urinary risk factors, to date, we cannot recommend it as a safe diet for kidney stone prevention.

### 3.6. The DASH Diet

The Dietary Approaches to Stop Hypertension (DASH diet) is a diet for which the connection with KSD was evaluated more thoroughly. This diet, rich in fruits and vegetables, with low-moderate intake of fat and proteins, respectively, had a reduced acid load compared to control (31 vs. 78 mEq/d of endogenous acid production), and it had a lower risk for stone formation [75,81]. Additionally, consuming a high number of fruits and vegetables and low-fat dairy products increased the urinary volume and urinary citrate, leading to a reduced risk of stone events by up to 45% [75,82,83]. This potentially protective effect was also echoed in the next diet, the Mediterranean diet, which differs from the DASH diet mainly by recommending a moderate intake of fat, instead of low or no-fat-only dairy products, as recommended by the DASH diet, while sharing a similar set of dietary recommendations regarding the high intake of fruit and vegetables.

### 3.7. The Mediterrenean Diet

The Mediterranean diet might play an important protective role against urinary stone formation. This was thoroughly demonstrated in a detailed study by Prieto et al., in which the calcium and uric acid urinary crystallization risks were calculated [84]. The authors found that a high consumption of vegetables was the strongest dietary factor for decreased calcium crystallization risk. However, for uric acid crystallization risk, it was a combination of low animal protein intake and healthy fatty acids intake, such as unsaturated fat from olive oil, for example. It should be noted, however, that the study population was composed of an elderly, mainly overweight population with metabolic syndrome. Nonetheless, given the close relationship between the metabolic syndrome and nephrolithiasis, the results are encouraging and might hint at a larger benefit for the wider population. However, this needs to be demonstrated in large heterogenic-cohort studies.

In conclusion, it seems that both the DASH and the Mediterranean diets exert a protective role mainly attributed to the lower intake of animal protein and high intake of fruits and vegetables.

### 3.8. Other Diet Types

Diets that are not modifications of the above-mentioned categories are usually diets comprising intermittent fasting periods. One example is the moon diet (also named the werewolf or lunar diet), which consists of short-term fasting periods of 24 h, on a monthly basis, during the full moon or new moon period [85]. In this diet, water intake is continued, and due to its short period, there is probably no change in urinary risk factors for KSD.

Detox diets are usually comprised of a short period of fasting, followed by a diet based on fruits and vegetables and fresh juices. These diets probably do not increase the risk for urinary stone formation, as long as water intake is continued, and calcium intake is maintained within the normal limits of 1–1.2 g per day.

## 4. Conclusions

Given the close relationship between KSD and obesity, it is surprising to find how limited the literature is when searching for the effects of diets to treat obesity, on the risk for KSD, as highlighted in the PRISMA flow chart. The drawing of clear conclusions for every diet is practically impossible, due to the variability of diet types, the presence of modifications of each diet that appear every few years, and the absence of long-term follow up studies. For this reason, this review studied the effects of each diet type on individual urinary risk factors for KSD. Nonetheless, the reduction of carbohydrate in the diet, while counterbalancing protein rich diets with sufficient intake of fruits and vegetables, seems to play a protective role against KSD formation. These conclusions need to be supported by additional research evaluating the direct relationship between KSD and each diet type, maintained for a sufficient time-period, to truly represent the change in urinary risk factors.

## Figures and Tables

**Figure 1 nutrients-13-04270-f001:**
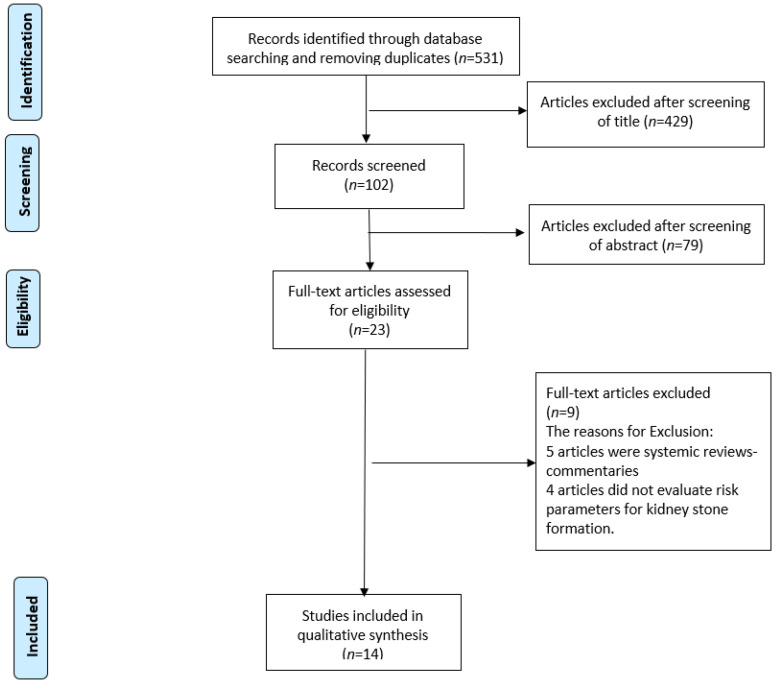
PRISMA flowchart of the included studies.

**Table 1 nutrients-13-04270-t001:** Summary of diets and potential effect on urinary parameters (Carb: Carbohydrates, CaOx: Calcium Oxalate).

Diet	Urinary Calcium	Urinary Oxalate	Urinary Uric Acid	Urinary Citrate	General Risk for Kidney Stones
Low Carbohydrate Diets - Atkins - Zone - Keto Diet - South-Beach	Mild decrease	Mild decrease	Decrease	——	Possible decreased risk. However, potential benefits of lower carb intake might be offset by increased protein intake.
Protein Rich Diets - Doucan - Paleo	Increase	Possible increase in CaOx Stone formers	Increase	Decrease	Possible Increased risk if protein intake exclusively from animals—can be counterbalanced by protein intake from vegetables.
Vegetarian Diets - DASH diets	No change if dairy product intake maintained	Possible increase with intake of high oxalate vegetables	——	Increase	Probable decreased risk under the condition of normal calcium intake and avoidance of high oxalate-content vegetables (e.g., Spinach, Rhubarb).
Vegan Diets	Decrease if dietary intake not maintained	Potential Increase due to low calcium intake	Potential Increase	Increase	Potential increased risk due to low calcium intake and resultant high urinary Oxalate content.

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
