# Peer review of "The Relationship between Modern Fad Diets and Kidney Stone Disease: A Systematic Review of Literature"

_nutrients, 2021, doi:10.3390/nu13124270_

Round 1
Reviewer 1 Report
Yazeed Barghouthy et al. propose a systematic review of literature on the relationship between modern fad diets and kidney stone disease. After identification of three principal diet patterns (manipulation of nutrient content in terms of low carbohydrate/high protein content), restriction of specific foods (vegan/vegetarian), intermittent fasting, they analyze literature data for risk factors liked to such dietary habits.
The authors approach the issue of the relationship between obesity, metabolic syndrome and nephrolithiasis from a different point of view than usual: not only obesity (and metabolic syndrome) represents a "direct" risk factor, but also soe dietary strategies can be.
The idea is interesting and the article is pleasant to read. Some points could be deepened to increase the interest in the paper.
The systematic review should be presented according to the PRISMA guidelines, above all because as the authors themselves say "". The flow chart could be to highlight even more the difficulty in finding scientific evidence of the impact of fad diet on the risk of nephrolithiasis (and, at the same time, highlight the inherent risk of using unconventional diets).
It would be useful, so that even a less experienced reader in the field of nephrolithiasis, could understand the importance of the question, to add to the introduction a part relating to the physiopathogenetic mechanisms of nefrolithiasis (for example the role of oxalate, uric acid, etc.).
It would also be interesting to mention the other risks associated with the use of fad diets, explaining the rationale for focusing the review on nephrolithiasis.
Author Response
Yazeed Barghouthy et al. propose a systematic review of literature on the relationship between modern fad diets and kidney stone disease. After identification of three principal diet patterns (manipulation of nutrient content in terms of low carbohydrate/high protein content), restriction of specific foods (vegan/vegetarian), intermittent fasting, they analyze literature data for risk factors liked to such dietary habits.
The authors approach the issue of the relationship between obesity, metabolic syndrome and nephrolithiasis from a different point of view than usual: not only obesity (and metabolic syndrome) represents a "direct" risk factor, but also soe dietary strategies can be.
The idea is interesting and the article is pleasant to read. Some points could be deepened to increase the interest in the paper.
Answer: We really thank the reviewer for summarising our paper well. We are grateful for their positive and supportive comments for our paper.
The systematic review should be presented according to the PRISMA guidelines, above all because as the authors themselves say "". The flow chart could be to highlight even more the difficulty in finding scientific evidence of the impact of fad diet on the risk of nephrolithiasis (and, at the same time, highlight the inherent risk of using unconventional diets).
Answer: Thank you for your comment, indeed this is very true and for this reason A PRISMA chart was added and a reference in the method and conclusion sections was made.
It would be useful, so that even a less experienced reader in the field of nephrolithiasis, could understand the importance of the question, to add to the introduction a part relating to the physiopathogenetic mechanisms of nefrolithiasis (for example the role of oxalate, uric acid, etc.).
Answer: Thank you for your comment, a paragraph in the introduction has now been added for this purpose. Line 38-51
It would also be interesting to mention the other risks associated with the use of fad diets, explaining the rationale for focusing the review on nephrolithiasis.
Answer: Thank you for your comment, a paragraph in the introduction was added for this purpose. Lines 77-85
Reviewer 2 Report
Manuscript number: nutrients-1456923
Title: The relationship between modern fad diets and Kidney stone 2 disease: A systematic review of literature
Authors: Barghouthy Y, Corrales M, Somani B.
Barghouthy and colleagues performed a systematic review of international literature to evaluate the lithogenic power of different fad diets.
The review appears in line with the aims and scope of the Journal. The manuscript is written in standard English but appears difficult to read.
However, despite the article is of interest, I have pointed out some criticisms that in my opinion significantly impair the study relevance.
A stringent definition of modern fad diet should be done to justify the selection of diets reported in the manuscript
A PRISMA diagram should be showed
Surprising, the anti-lithogenic power of Mediterranean diet is not analysed. If the authors do not consider the Mediterranean Diet as a modern fad diet, a comparison between each considered fad diet and the Mediterranean diet should be performed, considering the Mediterranean diet is considered by UNESCO a cultural heritage of the humanity
Author Response
Reviewer 2:
Barghouthy and colleagues performed a systematic review of international literature to evaluate the lithogenic power of different fad diets.
The review appears in line with the aims and scope of the Journal. The manuscript is written in standard English but appears difficult to read. However, despite the article is of interest, I have pointed out some criticisms that in my opinion significantly impair the study relevance.
Answer: We thank the reviewer for their positive comment to our paper.
A stringent definition of modern fad diet should be done to justify the selection of diets reported in the manuscript
Answer: Thank you for your comment, a paragraph in the introduction was added for this purpose lines 55-62.
A PRISMA diagram should be showed
Answer: Thank you for your comment, a PRISMA chart was added and a reference in the method and conclusion sections was made.
Surprising, the anti-lithogenic power of Mediterranean diet is not analysed. If the authors do not consider the Mediterranean Diet as a modern fad diet, a comparison between each considered fad diet and the Mediterranean diet should be performed, considering the Mediterranean diet is considered by UNESCO a cultural heritage of the humanity
Answer: Thank you for the accurate comment. We have added a comment regarding the anti lithogenic properties of the Mediterranean diet in the section 3.6 and 3.7.